# Mental Health Impacts of Tornadoes: A Systematic Review

**DOI:** 10.3390/ijerph192113747

**Published:** 2022-10-22

**Authors:** Sangwon Lee, Jennifer M. First

**Affiliations:** College of Social Work, University of Tennessee, Knoxville, TN 37996, USA

**Keywords:** tornadoes, mental health, systematic review, trauma, resilience

## Abstract

Tornadoes are one of the most prevalent natural hazards in the United States, yet they have been underrepresented in the disaster mental health comprehensive literature. In the current study, we systematically reviewed available scientific evidence within published research journals on tornadoes and mental health from 1994 to 2021. The electronic search strategy identified 384 potentially relevant articles. Of the 384 articles, 29 articles met the inclusion criteria, representing 27,534 participants. Four broad areas were identified: (i) Mental health impacts of tornadoes; (ii) Risk factors; (iii) Protective factors; and (iv) Mental health interventions. Overall, results showed adverse mental health symptoms (e.g., post-traumatic stress disorder, depression, anxiety) in both adult and pediatric populations. A number of risk factors were found to contribute to negative mental health, including demographics, tornado exposure, post-tornado stressors, and prior exposure to trauma. Protective factors found to contribute to positive outcomes included having access to physical, social, and psychological resources. Together, these findings can serve as an important resource for future mental health services in communities experiencing tornadoes.

## 1. Introduction

In the United States, there are more than 1200 tornadoes every year, contributing to more than 15,000 tornado-related fatalities since 1900 [1,2]. The average number of tornadoes has been increasing since 1954, and the likelihood of extreme tornadoes is also increasing [3], which could exacerbate the severity of the losses and negative effects of tornadoes in the future. While being directly exposed to a tornado can result in death, injuries, and physical damage to buildings and property, it can also have long-term effects on people’s mental health [4]. Emotional distress, anxiety, depression, post-traumatic stress, sleep disorder, and suicidal ideation have been reported as mental health problems experienced by victims after severe weather events [5,6,7]. These psychological symptoms can occur in the immediate aftermath of exposure and can persist over months to years [8]. In addition, people living with adverse mental health symptoms in post-extreme weather events reported a lower quality of life and functional impairments in social, vocational, and physical areas [9].

Compared to other types of frequent natural hazards (e.g., hurricanes, floods), comprehensive studies on the mental health impacts of tornadoes are lacking. For example, multiple systematic reviews have been conducted on the mental health impacts of hurricanes [10,11] and floods [12,13]. To the best of our knowledge, there have been no systematic reviews conducted and published on the mental health impacts of tornadoes. Therefore, we aimed to systematically identify and synthesize the available scientific literature on the mental health impacts of tornadoes. These findings can be used to (1) understand the mental health impacts of tornadoes, (2) identify possible risk and protective factors associated with psychological impacts, and (3) identify gaps in the literature to inform future research.

## 2. Materials and Methods

### 2.1. Literature Search

Using recommendations from the PRISMA group, we conducted a systematic review to identify and summarize findings of the peer-reviewed literature on tornadoes and mental health. We carried out literature searches for this topic through APA PsycInfo^®^, PUBMED, SCOPUS, and Web of Science. The search was restricted from January 1994 to December 2021 because after the Southeastern United States Palm Sunday Tornado Outbreak of 27 March 1994, there has been an increasing focus on empirical evidence-based research. We used free text, and words were restricted to title and abstract. The word “tornado” and “intervention” had an asterisk (*) added as a wildcard in to pick up plurals. The search strategy was:

Tornado* AND Psychological (Title/Abstract) OR “Mental health” (Title/Abstract) OR PTSD (Title/Abstract) OR “Posttraumatic disorder” (Title/Abstract) OR “Post-traumatic stress disorder” (Title/Abstract) OR PTSS (Title/Abstract) OR “Post-traumatic stress syndrome” OR Anxiety (Title/Abstract) OR Depression (Title/Abstract) OR “Disaster mental health” (Title/Abstract) OR “Intervention* AND Mental (Title/Abstract)”

### 2.2. Eligibility Criteria

All articles focused on the effects of tornadoes on mental health were included. Studies that explored the mental health impacts of tornadoes along with other disasters (e.g., tornadoes and hurricanes, tornadoes, and floods) were excluded. Articles that focused solely on the effects of tornadoes on the environment were also excluded. We included all articles dealing with: (i) mental health impact during and after tornado exposure; (ii) risk and protective factors that affect mental health during and after tornado exposure; (iii) evaluation tools and interventions to mitigate the impact of tornadoes on mental health. The following study designs were included in the systematic mapping (i) quantitative methods: survey, secondary data analysis, and data analysis after interviews; (ii) qualitative studies: individual interview; (iii) mixed methods: focus group interview (FGI) and survey, individual interview and survey. A protocol paper, case reports, letters, and editorials were excluded. We included only articles written in English.

### 2.3. Study Selection

Studies were screened for inclusion in two stages. The first author went over the title and published abstracts and made a preliminary list of articles in phase one. The authors assessed all articles published after 1994 and chose those that they thought fulfilled the inclusion criteria for full-text reading. Articles that were published before 1994 were excluded for two reasons: (i) to include the most recent literature; and (ii) to avoid possible inconsistencies in the description of mental disorders, as it was in 1994 that the 4th version of the Diagnostic and Statistical Manual of Mental Disorders was published [10]. When there was a disagreement, the researchers discussed their different viewpoints and came to an agreement. The authors assessed the entire publications that matched the eligibility criteria in the second phase.

### 2.4. Data Extraction

The first author extracted essential features from journals that met the inclusion criterion, such as methods, characteristics, and results. The co-author reviewed the essential features, study design, demographics, and a brief overview of the findings. Based on the key features and a summary of studies, the authors identified four broad categories of the topic.

### 2.5. Assessment of the Risk of Bias

To assess for bias in the studies, the Hoy Risk of Bias Tool (RoBT) [14] was used. The RoBT consists of ten items evaluating external (four items) and internal (six items) validity. In this review, we adopted eight items of the tool. Two of the six questions for evaluating internal validity were not adopted because they were not applicable to evaluating the articles reviewed in this paper. Studies were classified as having a low risk of study bias when six or seven of the eight items were answered as “yes (low risk),” and moderate risk of bias when three to five of the questions were answered as “yes (low risk),” and a high risk of bias when zero to two questions were answered as “yes (low risk)”. Seventeen studies had a rating of low bias, eleven studies had a rating of moderate bias, and one had a high-risk rating. A result of the assessment of the risk of bias is available in Appendix A Supporting Information.

## 3. Results

### 3.1. Literature Search and Study Selection

The electronic search strategy identified 384 potentially relevant articles. After removing duplicates and irrelevant, or those that were published before January 1994, 61 were reviewed by reading the title or abstract. We assessed 35 full-text journals for eligibility, and 29 were included for further review (Figure 1). A summary of the journals selected is available in Appendix A Supporting Information.

### 3.2. Characteristics of the Studies Selected

We found that 23 of the articles (79.3%) used quantitative methods. Of these, 21 were surveys. Three (10.4%) were qualitative studies (individual interviews) dealing with the subjective experiences of tornado exposure. Three (10.4%) articles were mixed methods of survey and focus group interviews or individual interviews. Regarding the demographics of the articles, 22 articles (75.9%) focused on children, adolescents, and their caregivers. Five (17.2%) focused on adults, and three (10.4%) focused on the general population. However, we were not able to find research that focuses on the elderly population. Twenty-one articles (72.4%) were focused on the Missouri and Alabama tornado outbreak in 2011, and five (17.2%) were focused on tornadoes in Oklahoma. One was focused on the Minnesota tornado in 1998. One study that focused on the tornado in Jiangsu, China, in 2016, was found. (See Table 1).

### 3.3. Main Outcomes

We analyzed 29 articles, representing 27,534 participants. Four broad areas were identified: (i) Mental health impacts of tornadoes; (ii) Risk factors; (iii) Protective factors; and (ⅳ) Mental health interventions
(i)Mental health impacts of tornadoes: In this section, we present several mental health consequences reported to be associated with exposure to tornadoes.
Post-traumatic stress disorder (PTSD): We found in multiple studies that residents exposed to tornadoes reported PTSD or PTSD-related symptoms. Degree of tornado exposure, gender, age, emotional support, and barriers to a tornado warning were factors related to post-tornado PTSD [4,15,17,19,21,25,26,32,35,38,39]. PTSD, or symptoms associated with PTSD, was also found to be related to other mental and behavioral health needs such as depression [20,21,25], binge drinking [16], or substance abuse [15]. Studies found women were more likely to report more PTSD symptoms than men after a tornado [4,21,35,39], and one study found men were more likely to report depression symptoms following the Joplin 2011 tornado [25]. The questionnaires to measure PTSD were a modified version of the SF-12, The National Survey of Adolescents Replication PTSD Modules, OSU PTSD Scale-CF, the Post-traumatic Stress Disorder Checklist for Civilians, the Trauma Screening Questionnaire, Frederick Reaction Index, and the Child PTSD Symptom Scale and The UCLA PTSD index.Depression: We found that depression was also a common mental health consequence experienced months after the tornado [20,21,38], but also 2.5 years later [25]. Adolescents who reported two or more depressive symptoms had 3.5-fold odds of increased risk of PTSD [21]. In addition, low education, low social support, and tornado exposure increased the likelihood of a diagnosis of depression [20,21,25]. Questionnaires used to assess depression included a revised version of SF-12, the Patient Health Questionnaire-2 Depression Subscale, the Patient Health Questionnaire-8, The National Survey of Adolescents-Replication PTSD and MDD Modules, and the NSA-Depression module.Anxiety: Several studies reported an elevated prevalence of generalized anxiety disorder following tornadoes. In six out of 29 studies [22,28,38,39,40,42], the level of anxiety was measured using the modified Differential Sentiment Scale, the OSU PTSD Scale, or was measured through an interview. Anxiety experienced by residents after the tornado was also thought to be a contributing factor to PTSD symptoms [22,39]. For children, the degree of exposure to the tornado was found to be associated with anxiety level [28]. Measures to assess anxiety in children included the Generalized Anxiety Disorder-7.Alcohol and substance abuse: Studies examining increased alcohol consumption in adolescents after tornadoes were found to be significantly influenced by previous trauma history and current levels of tornado-related PTSD [16,31]. We found conflicting results on whether alcohol abuse was predicted by age and gender. One study reported higher levels of alcohol consumption among men and older adolescents [16]. However, in another study, gender was not a significant predictor of binge drinking, nor did it act as a risk factor for increased alcohol use after a tornado [31]. Rutgers Alcohol Problem Index and Quantity-Frequency Index of Alcohol Consumption questionnaires were used to identify problems experienced as a result of drinking. Substance use disorder (SUD) was found to be associated with post-traumatic stress disorder and major depressive episodes following exposure to tornadoes. Girls were significantly more likely than boys to meet the diagnosis criteria of comorbidity for “PTSD and major depressive episode”, and “major depressive episode and substance use disorder”. Adolescent SUD was assessed using the CRAFFT screening test [15].Suicidal ideation: About 5% of 2,000 adolescents who experienced the Joplin and Tuscaloosa tornadoes in the spring of 2011 reported suicidal thoughts [33]. Intimate partner violence (IPV) exposure was also found to be significantly related to post-tornado suicidal ideation, even after accounting for current mental health symptoms (i.e., PTSD and depression). Notably, prior disaster exposure and demographic characteristics were not significantly associated with suicidal ideation, suggesting that certain factors of IPV may predict suicide risk.Children’s emotions and distress: Avoidance, re-experiencing, interpersonal alienation, interference with daily functioning, physical symptoms/anxiety, and foreshortened future were found to be elements of mental health impacts of tornadoes upon children [37]. Studies found that emotional distress (i.e., fear) and the amount of damage to schools were also associated with overall psychological effects on children. In children, a combination of attributions, particularly meaning-seeking and perceived tornado exposure, were overall predictors of long-term distress after a tornado [27]. In addition, children’s emotional processing was found to impact children’s meaning-making efforts and have implications for their adjustment after a tornado [23,41]. Finally, maternal support was found to be important in the relationship between children’s use of both positive and negative emotional language and child tornado-related post-traumatic stress syndrome (PTSS). Maternal support included asking their children questions, making follow-up statements, repeating back content by paraphrasing, and providing possible solutions to problems [36].(ii)Risk factors: In this section, we present risk factors that were found to contribute to adverse mental health after a tornado. Although we have grouped each factor into categories to aid the reader’s understanding, individual factors and social risk factors are systematic and closely related and may be addressed comprehensively in research.
Gender and age: Multiple studies found females with low levels of education and lower household income were more vulnerable to adverse mental health after tornado events [25,38,39]. Children of parents who suffer from adverse mental health conditions (e.g., depression) after a disaster often experience similar difficulties [24,29]. In examining the relationship between age and PTSD in children and adolescents, we found two conflicting results. In one study, young children between the ages of 4 and 10 were more likely to experience PTSD after a tornado [24], while another study found that older adolescents were more likely to report symptoms of PTSD [29]. For boys, the effect of tornado exposure on PTSD and depression increased as social support decreased. However, for girls, regardless of the level of tornado exposure, social support was associated with PTSD and depression [29].Race and ethnicity: Two articles addressed specific challenges that the African-American and Latinx populations faced in the context of tornadoes [4,39]. Black and Latinx residents who reported difficulties receiving tornado warning alerts experienced more tornado exposure and adverse mental health impacts, and the intersections of ethnicity, language difficulties, citizenship, pre-mental health symptoms, and social class were also found to contribute to additional stressors [4].Economic factors: Having lower income, renting, having debt, not having insurance, and loss of material resources were the significant risk factors for post-tornado adverse mental health symptoms [19,24,25,29,39]. However, there was one contradictory finding that, during long-term recovery from the Oklahoma Moore Tornado in 2013, loss of personal characteristics such as self-esteem or confidence were all statistically significantly associated with adverse mental health, whereas material loss did not correlate with mental health outcomes [19].Tornado exposure: Multiple studies have identified that tornado exposure (e.g., injury, losing a loved one, property damage) is a strong factor that is linked to negative mental health for children, youth, and adults [15,17,19,22,24,25,27,28,29,31]. Families who experienced greater severity of tornado exposure had youth who reported more PTSD and caregivers who reported more distress [15,17]. Greater severity of tornado exposure was also associated with alcohol use by adolescents [16,31]. However, two studies found that tornado exposure was unrelated to child-reported emotions and mental health symptoms after a tornado [23,41]. Tornado Exposure Questionnaire, The Tornado-Related Traumatic Experiences Questionnaire, and Natural Disaster Experiences Inventory were used to measure the severity of tornado exposure.Pre-tornado trauma experience: Past trauma exposure (e.g., prior exposure to natural hazards, prior exposure to accidents) and IPV were risk factors for post-tornado mental health symptoms [30]. Among adolescents, prior exposure to traumatic events was the most consistent predictor of each post-tornado comorbidity (comorbid post-traumatic stress disorder, major depressive episode, and substance use disorder) profile [15]. Females and adolescents with an IPV history in addition to a natural hazard experience, had increased PTSD symptoms [30,35].Lack of mental health services: Finally, one study examined accessing mental health services in the 2.5 years following the 2011 Joplin, Missouri tornado [25]. 2.5 years after the tornado, the majority of respondents reported they had not accessed mental health support, and had little or no contact with a counselor or other mental health professional (83.4%) or religious leader (85.9%).(iii)Protective factors: In this section, we present protective factors related to individuals’ coping or adaptation and resilience after tornadoes.
Coping response factors: Coping responses found to have a positive impact on personal recovery and mental health included dispositional optimism, self-efficacy, and hope. Such coping responses were found to moderate the association between the severity of home damage and personal recovery, as well as the relationship between home damage and PTSD [18]. Another study found that individuals’ religiosity and emotional coping in response to tornadoes predicted taking protective action during tornadoes. Related to emotions, anxiety and fear were found to contribute to protective decision-making (e.g., sheltering in place or collecting supplies), which was then related to positive mental health outcomes [40].Social support: Multiple studies found that social resources were positively associated with higher levels of resilience and negatively related to adverse mental health outcomes following tornadoes [16,24,32,37]. One study that focused on disaster interpersonal communication and post-traumatic stress following the 2011 Joplin, Missouri tornado found that post-traumatic stress symptoms (PTS) were associated with more frequent communication with family, friends, and neighbors about the tornado [24]. Another study found that environments with fewer social resources and support influenced the risk of alcohol consumption and alcohol use disorders of residents [16]. Finally, perceived social support was found to reduce PTSD symptoms and increase Post-traumatic Growth (PTG) among participants affected by the tornado in Yancheng, Jiangsu, China [32].(iv)Mental health interventions: In this section, we detail interventions found to support mental health following exposure to tornadoes. We found two mental health interventions for children and adolescents, and both were utilized after the 2011 tornado event in Joplin, Missouri.
Journey of Hope: “Journey of Hope” is a program designed for young people exposed to disasters who do not meet the diagnostic criteria for formal mental health symptoms but experience emotional distress. This intervention uses a school-based psychosocial curriculum that includes eight sessions aimed at improving the emotional management of children and adolescents by improving protective parameters such as social support, coping, and psychological education. We found one study that explains 110 participants ranging from 11 and 15 years old who experienced Moore, Oklahoma tornado in 2013 improved their ability to prosocial behaviors after completing the program. According to the study, a statistically significant increase in the Strengths and Difficulties Questionnaire subscale for prosocial behaviors was found following the completion of the intervention [42].Bounce Back Now: “Bounce Back Now (BBN)” is a web-based intervention and consists of five standalone modules that intervene with PTSD, depression (mood), smoking, alcohol use, and parenting for adolescents at risk for post-disaster mental health problems. We found a population-based randomized controlled trial of BBN. Two thousand adolescents and parents who were affected by the tornado outbreak in Joplin and several areas in Alabama enrolled in the intervention, and nearly half of the disaster-affected families accessed the intervention. Youth and their parents were randomly assigned to (a) a web intervention for disaster-affected youth, (b) a web intervention for disaster-affected families (youth and parents), and (c) an evaluation-only web comparative experiment. Researchers found fewer PTSD and depressive symptoms for adolescents in the experimental versus control conditions at a 12-month follow-up. Results revealed the feasibility and initial efficacy of BBN as a scalable post-disaster mental health intervention for adolescents [34].

## 4. Discussion

To the best of our knowledge, this is the first systematic review that focuses on reviewing and synthesizing the literature on the mental health impact of tornadoes. After reviewing a total of 29 papers representing 27,534 participants, we identified four broad areas: mental health impacts of tornadoes, risk factors, and protective factors and mental health interventions. In terms of mental health impacts, we found tornadoes can have negative effects on mental health, including PTSD, anxiety, depression, suicidal thoughts, and alcohol and drug abuse in children, adolescents, and adults. Individual factors such as gender, age, ethnicity, economic status, exposure to tornadoes, pre-tornado traumatic experiences, and the social context that prevents post-disaster victims from accessing mental health services were all found to be risk factors for post-tornado mental health problems. We found that women, children, and adolescents have a relatively higher incidence of PTSD and mental health symptoms than men and adults after tornadoes. These results are the same in studies of other natural hazards such as hurricanes, earthquakes, and tsunamis [43,44]. In addition, we found two articles that focused on Black and Latino populations. The interconnections of their race, citizenship, legal issues, language barriers, and socioeconomic status were major stressors that needed to be addressed in post-disaster services. Other risk factors found included economic factors such as having lower income, renting, having debt, not having insurance, and loss of material resources were the risk factors for post-tornado adverse mental health problems. Future research is warranted to examine various sociocultural dimensions of risk from an intersectional framework (e.g., race and ethnicity, national origin, class, physical ability, age, and living in a rural or urban environment).

In terms of protective factors, individuals’ religiosity, hope, and dispositional optimism were found to contribute to better mental health outcomes and personal recovery. In addition, material and social resources were found to serve as important protective factors. These combined findings highlight the importance of both internal (e.g., hope, optimism) and external (e.g., material resources, social support) protective factors contributing to better mental health outcomes following tornadoes. These findings can assist practitioners in identifying protective factors that guide a framework for interventions and practice models that build resilience in post-tornado settings.

We also identified two practical interventions aimed at relieving distress and emotions in children after a tornado. In the context of natural hazards, children and adolescents are characterized as one of the most vulnerable and highly dependent [45,46]. In addition, children and adolescents who have experienced a disaster may experience depression and post-traumatic symptoms long after the event [47,48]. Childhood trauma survivors present alcohol and drug dependency issues, and early onset of trauma may contrive adverse mental health symptoms such as anxiety in adulthood [49]. For these reasons, we conclude that further research into long-term recovery programs that can develop social-emotional skills in children and adolescents may be necessary, along with follow-up studies of “Bounce Back Now” and “Journey of Hope”.

Based on these findings, we identified some important research gaps. In terms of methodological aspect, of the total 29 studies, 23 studies were conducted based on a quantitative methodology. In future research, it may be required to investigate the potential for a meta-analysis to provide additional support for the findings of individual quantitative studies. We also found that the majority of studies focused on children, adolescents, and caregivers. Notably, there have been no studies on mental health after tornadoes experienced by the elderly population and people with disabilities in the past 27 years. The paradox that older people and people with disabilities are at greater risk of disasters [50,51,52] but are excluded from disaster studies strongly suggests that studies involving these populations are needed. Given the increasing likelihood of tornadoes due to climate change [53] and a growing older population worldwide [54], it is important for policymakers and practitioners to have evidence of the impact of tornadoes on the mental health of aging populations. Furthermore, we found that the LGBTQ population was also essentially invisible following tornadoes as there was no article that focused on this group. LGBTQ people are more likely to experience social isolation and to endure disrespect or harassment in places like emergency shelters in the aftermath of a disaster [55]. Therefore, future research may be needed to explore the social isolation and consequent mental health problems experienced by this group after tornadoes. In addition, experiencing more tornado exposure (e.g., being injured, losing a loved one, housing damage) was shown to be associated with risk factors for adverse mental health, including studies from above. Prior studies indicate that individuals living/sheltering in mobile/manufactured homes have a greater probability of tornado-related injuries, fatalities, and losses [56,57]. Future research is warranted to examine mental health outcomes among mobile/manufactured home residents exposed to tornadoes.

Finally, we found most of the studies included in this review were published in the United States and focused on EF4 or EF5 tornadoes, with many studies being related to the Missouri and Alabama tornado outbreak in 2011. Severe weather has increased over the past few decades with climate change [53], and the geographic areas with high frequencies of tornado activity are gradually shifting to the U.S. southeast region [58,59]. Therefore, we recommend additional studies on tornado-related mental health outcomes from populations in the southeast. We also recommend future studies examining tornado-related mental health needs in communities with low-attention tornadoes (EF 1-3) that may not warrant federal or state support but may have considerable losses and mental health needs.

In this review, we summarized the peer-reviewed literature examining mental health impacts during and after tornado events. Although the findings of this study contribute to a better understanding of the impact of tornadoes on mental health, there are certain limitations that should be addressed in future research. First, although we included the most significant mental health consequences and keywords associated with disaster mental health in our search method, such as depression, anxiety, PTSD, mental, and psychological, it is probable that some research papers that identify other outcomes were missed. Second, we only searched articles that were written in English. In future research, if we search for papers written in multiple languages, we will be able to find data on tornadoes outside the United States and their aftermath. This attempt will add diversity and depth to tornado research, which is lacking in quantitative and qualitative compared to other natural hazards research.

## 5. Conclusions

We systematically reviewed the available scientific evidence on the mental health effects of tornadoes within published research journals. We extracted data from 29 articles that met the inclusion criteria, and four broad areas were identified: (i) Mental health impacts of tornadoes; (ii) Risk factors; (iii) Protective factors; and (iv) Mental health interventions. These findings indicate that tornadoes can have a significant impact on mental health in communities exposed (e.g., post-traumatic stress disorder, depression, anxiety) from months to years after a tornado. Studies have found increases in adverse mental health in both adult and pediatric populations. A number of risk factors were found to contribute to negative mental health, specifically demographics, post-tornado stressors, and prior exposure to trauma. Furthermore, a variety of protective factors were found to increase positive mental health outcomes. These included having access to physical, social, and psychological resources. Together, these findings can serve as an important resource for future mental health interventions in communities experiencing tornadoes. We also identified a need for future post-tornado mental health research among diverse and socially disadvantaged populations, including older adults, mobile/manufactured residents, people with disabilities, and the LGBTQ populations. Future studies examining these areas can help to identify factors and resources to protect individuals from negative consequences after tornado-related adversity and further develop disaster resilience systems.

## Figures and Tables

**Figure 1 ijerph-19-13747-f001:**
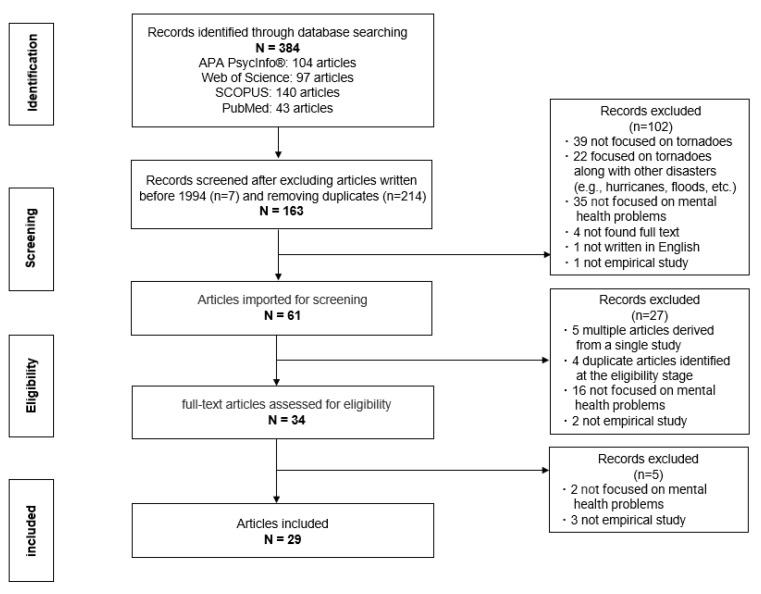
Flow diagram.

**Table 1 ijerph-19-13747-t001:** Summary of the characteristics of included references.

Types of Study		No. in References	N	%	Total No. of Participants
Quantitative methods	Survey	[4,15,16,17,18,19,20,21,22,23,24,25,26,27,28,29,30,31,32,33,34]	21	72.4	23,260
	Secondary data analysis	[35]	1	3.4	2000
	Data analytic after interviews	[36]	1	3.4	118
Qualitative methods	Individual interview	[37,38,39]	3	10.4	435
Mixed methods	FGI and survey	[40]	1	3.4	1561
	Individual interview and Survey	[41,42]	2	6.9	160
Characteristics of the Sample					
Age	Children	[22,26,28,41]	4	13.8	614
	Children and parents (or caregivers)	[23,27,36]	3	10.4	2396
	Adolescents	[15,20,31,32,33,35,39]	7	24.1	6965
	Adolescents and parents (or caregivers)	[16,17,21,29,30,34,37]	7	24.1	11,515
	Adults	[4,18,24,25,40]	5	17.2	5782
	General	[19,38,39]	3	10.4	262
Location, year, and EF scale of Tornadoes					
The U.S.	Missouri and Alabama (2011)	[15,16,17,18,20,21,23,24,25,28,29,30,33,34,35,36,37,38,39,41]	20	69.2	22,922
	Oklahoma (2001)	[27]	1	3.4	198
	Oklahoma (2006)	[22,26]	2	6.9	204
	Oklahoma (2013)	[19,42]	2	6.9	181
	Middle Tennessee (2020)	[4]	1	3.4	221
	Minnesota (1998)	[31]	1	3.4	2000
	Non-specific	[40]	1	3.4	1561
Outside of the U.S.	China (2016)	[32]	1	3.4	247
EF scale	EF5	[15,16,17,18,19,20,21,22,23,24,25,29,30,33,34,35,36,38,41,42]	20	69.2	24,488
	EF4	[4,28,31,32,37,39]	6	20.7	1235
	EF3	[27]	1	3.4	198
	Not specified	[26,40]	2	6.9	1613

## Data Availability

These data used to support the findings of this study are included within the article.

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
