# Peer review of "Mental Health Impacts of Tornadoes: A Systematic Review"

_ijerph, 2022, doi:10.3390/ijerph192113747_

Round 1

Reviewer 1 Report

Reviewer report 19 September 2022

Summary

This systematic review reported on a) the mental health impacts of tornadoes on affected communities, b) risk and protective factors that are associated with adverse mental health during and after a tornado and c) mental health interventions implemented to support young people affected by tornadoes. This paper appears to be the first to review this literature, and the findings indicate that tornadoes present a substantial risk to the mental health of adult, adolescent and child populations. The review also finds clear gaps in the literature (e.g. no studies on the mental health of older people affected by tornadoes), and highlights areas where risk factors may be modified and protective factors strengthened. Overall the paper provides a broad assessment of the current state of the literature in this area.

General concept comments

Introduction

·         Lines 36-38 - “…research on the mental health consequences of tornadoes has been neglected and is less thorough than that on other types of disasters (e.g., hurricanes, floods).” Please elaborate on this statement to strengthen the argument:

o   How has research in this area been neglected, e.g. is there limited funding for such research? Are there comparatively more papers on the mental health effects of hurricanes than tornadoes?

o   In what way is the research in this area less thorough, e.g. are the studies of poor methodological quality?

·         Lines 39-43 – This paragraph would benefit from a sentence which explains why the authors chose to conduct this review, e.g. because to date there have been no reviews published in this area.

Methods

·         Line 55 – were other variations of PTSD (e.g. spelling out post-traumatic stress disorder, and including acute stress disorder and post-traumatic stress syndrome) also included in the search? If not, please explain the rationale for this choice.

·         Lines 72-74 – “If two or more articles derived from a single study were identified, only the earliest published articles were included.” Please provide a rationale for this decision, given that a) it is implied in the Introduction that this area has received little attention to date and b) not including multiple articles based on the same study appears to weaken the assertions made about the effectiveness of the mental health interventions reported later in the manuscript.

·         Line 101 – “Three (10.2%) reviews were mixed methods…” Were review studies eligible for inclusion in this review? If so, this needs to be clarified in section 2.3.

·         Missing – the PRISMA guidelines referenced in line 43 recommend conducting a risk of bias assessment for studies included in a systematic review. Please include this information, justify why PRISMA guidelines were not followed, or remove the reference to the PRISMA guidelines.

Results

·         Table S1 – A lot of information in the Results and Discussion/Conclusion columns of the table is a direct copy/paste of the information from the original papers. It would be prudent for the authors to reword the information presented in the table and report quantitative results (e.g. percentages, statistical significance) wherever possible. This would supplement the summary of results presented in the body of the manuscript rather than reiterate it.

·         Consider removing some of the acronyms related to the symptom assessment scales from this section as they are only used once in the manuscript. Alternatively, the lists of screening tools used to assess different symptoms or phenomena could be taken out of the text and Table S1 and presented as a separate table; this would make the results section easier to read and more clearly answer eligibility criterion 3 (lines 75-76).

·         Lines 262-279 – The section on mental health interventions should stand alone, rather than being subsumed under protective factors. This is because a) the Methods section suggests that it will be a separate area of investigation (lines 75-76), b) protective factors are in place prior to adverse events, while the mental health interventions listed in this section were implemented after tornadoes occurred and c) there does not appear to be enough evidence to suggest that these interventions are protective against future tornado-related mental health effects.

Specific comments

Introduction

·         Lines 27-28 – “In the United States, more than 1,200 tornadoes per year, and since 1900 over 15,000 lives have been lost in tornadoes.” This is a clunky sentence. Are there some words missing?

·         Lines 25-38 – The paragraph often reiterates how common tornadoes are (lines 27, 29, 30, 35). Some of these statements could be removed, as the point only needs to be made once, and the space can be better used to strengthen the main argument, e.g. by outlining the results of the studies cited in this paragraph that find adverse mental health impacts in tornado-affected communities.

·         Lines 40-42 – “These findings can help to develop future recommendations that aid in improving the research and intervention not only on individuals' mental health but response and services of communities exposed to tornadoes.” This sentence contains several ideas and is hard to follow. It could be split into its component ideas, which could then be briefly described to give the reader a clearer understanding of how this review benefits the field, e.g. the results could be used to advocate for increasing the quality or quantity of future research into the mental health impacts of tornadoes to address known gaps in the literature.

Methods

·         Lines 57-59 – “Additional studies not found in the original database search were verified in the citation indices and reference lists of retrieved journals.” This statement is unclear. Do the authors mean that they hand-searched the reference lists of retrieved journal articles for eligible studies that were not captured by the original search?

·         Lines 60 and 68 (sections 2.2 and 2.3) – I suggest swapping the sections on eligibility criteria and study selection around, as eligibility criteria would need to be set before study selection took place.

·         Lines 70-71 – “Journals that explored mental health in the aftermath including tornados and other disasters together (e.g., hurricanes, floods) were excluded.” Unclear sentence - are there some words missing?

·         Line 74 – does “journals” refer to “journal articles”? Please check the rest of the manuscript and amend where necessary, e.g. lines 83, 92, 93, Figure 1.

·         Lines 74-75 – Criteria 1 and 2 should specify that this review includes articles that address mental health impacts/risk and protective factors both during and after tornado exposure. There are parts of the Results and Discussion sections where the language jumps around and affects the reader’s interpretation of the text.

·         Lines 79-80 – Please define the acronym FGI as this is the first time it is used in the manuscript. “Survey” does not need to be capitalised.

·         Figure 1:

o   Please clarify the meaning of “type of study” and “languages” in the first box on the right side of the diagram

o   Please clarify the meaning of “duplicates” in the second box on the right side of the diagram – does this refer to multiple articles derived from a single study, or were more duplicate records identified at the eligibility stage?

o   Please clarify the meaning of “irrelevant” in the second box on the right side of the diagram

Results

·         Lines 144-145 - “the Generalized Anxiety Disorder (GAD-7) used to evaluate anxiety.” Incomplete sentence.

·         Line 156 – “girls were more likely to be included criteria of diagnosis…”. Unclear.

·         Line 175 – Please define the acronym PTSS as this is the first time it is used in the manuscript

·         Lines 189-190 – “Adolescents who reported symptoms of depression were found to be 3.5 times more likely to be diagnosed with PTSD” – repeats lines 128-130 and appears out of place in this section. Suggest deleting.

·         Lines 190-192 – “In one study to explain the relationship between disaster exposure and post-traumatic stress symptoms (PTSS) of adolescents, researchers found adolescents' egocentrism was predictive of higher PTSS [20].” This sentence is not directly relevant to the associations between demographic characteristics and vulnerability to adverse mental health after a tornado. This sentence may be better placed somewhere else, or deleted.

·         Lines 199-201 – “Multi-dimensional exploration of barriers to tornado response and victims' legal status and social context caused by race could advance more efficient mental health interventions and tornado response systems.” Does this sentence report a result of the study being described in this paragraph, or is it a point for the discussion of this manuscript? The same comments applies to lines 232-234.

·         Lines 223-226 – “Among adolescent, prior exposure to traumatic events was the most consistent predictor of each post-tornado comorbidity (comorbid posttraumatic stress disorder, major depressive episode, and substance use disorder) profile [9].” This sentence is difficult to follow. It may be simpler to say “Among adolescents, prior exposure to traumatic events was the most consistent predictor of each post-tornado comorbidity profile [9].”

·         Lines 231-232 – percentages are reported to 2 decimal places here, but 1 decimal place in other places in the manuscript. Please be consistent in reporting percentages. For the same sentence, please specify the time period the reported results relate to, e.g. 2.5 years after the tornado, the majority of respondents had not accessed mental health support.

·         Line 248 – Please define the acronym PTS as this is the first time it is used in the manuscript

·         Lines 268-279 – The description of the BBN intervention and study results could be more informative, e.g. it could briefly explain how the intervention was delivered and how many sessions are involved. The information about completers vs non-completers should be removed as it does not add to the findings of the current manuscript, and should be replaced with a statement which notes that the cited paper doesn’t report the full results of the RCT, so the effectiveness of the program and possible future applications are currently unclear. However, BBN does seem to have been evaluated in study #17, so perhaps some of the results from that study could be reported in this section?

Discussion

·         As an overall comment, this section reiterates a lot of information from the Results section which could be deleted or condensed to make space for a more nuanced discussion of gaps in the literature and provide clearer directions for future research (e.g. the paragraph from lines 314-326 does this well). For example:

o   Lines 302-313 could be condensed to say that most studies in the review were published in the United States and studied populations affected by EF5 or EF4-rated tornadoes, and that there may be reason for future research to investigate populations from different geographic regions and groups who are affected by tornadoes of lesser severity.  

o   Lines 344-355 could be condensed and include concrete examples to better illustrate the points made, e.g. describing how protective factors could be promoted in communities vulnerable to tornadoes or adverse mental health (e.g. through educational campaigns) and how better communication could be incorporated into response planning (e.g. by better publicising the community’s disaster response plan)

o   Lines 395-399 could include a clearer explanation of how the results can inform tornado response systems and disaster planning for tornadoes, e.g. whether (more) mental health professionals should be sent to tornado-affected communities, and how soon or how often after a tornado

·         Lines 287-289 – “We inferred the reason for this phenomenon is the event of the Southeastern United States Palm Sunday Tornado Outbreak of March 27, 1994, and the revision of the Diagnostic and Statistical Manual of Mental Disorders (DSM–IV).” This sentence reads as highly speculative. Suggest softening the language (e.g. “This may be due to the Southeastern…”) and providing a clear rationale for why the publication of the DSV-IV could be linked to an increase in research in this area.

·         Lines 293-297 – “Individual factors such as gender, age, ethnicity, economic status, exposure to tornadoes and pre-tornado traumatic experiences, and the social context that prevents post-disaster victims from accessing mental health services, have been turned out to act as risk factors for exacerbating post-tornado mental health problems.” This sentence could be clearer, e.g. “Individual factors such as gender, age, ethnicity, economic status, exposure to tornadoes and pre-tornado traumatic experiences, and the social context that prevents post-disaster victims from accessing mental health services, may be risk factors for post-tornado mental health problems.”

·         Lines 300-301 – “Also, we identified two practical interventions that were found to mitigate children's distress and emotion after the 2011 tornadoes.” This sentence overstates the results. One of the studies evaluating one of the interventions found positive results, while the preliminary results of the other study showed no clear evidence of effectiveness.

·         Lines 304-306 – These sentences are contradictory. Was there one Missouri Alabama tornado, or many? Please also check the rest of the manuscript for consistency in reporting this event.

·         Lines 312-313 – “…we recommend that research on tornadoes in different regions and on various scales in future studies.” Please clarify this part of the sentence.

·         Line 324 – should read “these populations” rather than “this population”

·         Lines 342-347 – Some of the language is unclear in these sentences, e.g. recovery is not a concept that applies during an event, there may be a conflation of the results of existing studies (e.g. study #24 which examines protective actions taken during a tornado) with future research directions (e.g. how to implement effective crisis management in the aftermath of a tornado). Please review and clarify.

·         Lines 356-359 – As with earlier comments, suggest moderating the language used here, given that only two studies of these interventions were included. Additionally, as per the comments related to lines 72-74 above, it’s not clear why multiple papers evaluating these interventions were not included in this review; this may have strengthened the evidence base for these interventions and, subsequently, the claims made in this manuscript.

·         Line 365 – replace “JoH” with “Journey of Hope” for consistency across the manuscript.

·         Line 368-369 - “In future research, it may be required to try to grasp the lived experiences of tornado victims by using qualitative and mixed method.” It is not clear what more this would add to the literature, given that there appears to be a lot of evidence for the adverse effects of tornadoes on mental health outcomes, and a range of risk factors that may contribute to poor mental health after a tornado. A more logical next step might be to investigate the potential for a meta-analysis to provide additional support for the findings of individual quantitative studies.

·         Line 373 – This review spanned 27 years of literature, not 20.

·         Line 375 – should read “these populations” rather than “this population”

·         Line 382 – Citation #47 should reference the relevant FEMA National Preparedness Report directly, rather than a news article that comments on the report.

·         Line 387 appears to be missing some references.

Conclusions

·         This section reiterates quite a lot of information from the Results and Discussion. Consider deleting or condensing parts of this section so that the reader is left with a couple of clear messages about how this research contributes to the field and promising future research directions.

References

Please carefully check the article and journal titles in the reference list and ensure they conform to the journal’s standard – some words are capitalised when they shouldn’t be (e.g. Factors, line 455), and others are not capitalised when they should be (e.g. Social science & medicine, line 472). Other references seem incorrect, e.g. “LGBTQ People Are at Higher Risk in Disasters in Scientific America” (line 562).

Reviewer 2 Report

Thank you for the opportunity to review the manuscript titled "Mental health impacts of tornadoes: A systematic review."

Please, find below some comments, suggestions, and questions.

Line 27: the phrase "more than 1,200 tornadoes" seems incomplete

Line 56: Were the keywords Tornado* AND [a series of keywords, including mental health" AND "Mental"?

Line 58: were these additional studies verified or identified in the citation indices and reference lists?

Line 63: "were excluded to have access to the most recent research". I don't think there is a need to add another explanation as to why studies published before 1994 were excluded. The previous section explains that these were excluded "because after the Southeastern United States Palm Sunday Tornado Outbreak of March 27, 1994, there has been an increasing focus on empirical evidence-based research". This explanation was also put forward by Fernandez et al., (2015) "Flooding and mental health: a systematic mapping review." Please, acknowledge the source.

Line 74 "we included all journals dealing with." Did the authors mean to say articles instead of journals?

Line 114: "we have found psychological symptoms such as PTSD, depression and anxiety." I recommend rephrasing "mental health disorders and psychological symptoms such as" (PTSD is a mental health disorder, not a symptom).

Line 117: "found a prevalence of PTSD." Was this an increased prevalence? PTSD may be found in the population even when there are no tornadoes. Did these studies assess the prevalence of PTSD after a tornado?

Line 119: Was a particular gender or age associated with a higher prevalence of PTSD? It's easy to assume that lack of emotional support, barriers to a tornado warning, and a higher degree of tornado exposure increased the prevalence. Was this the case?

Note: it is explained in line 132 that women were more likely to be diagnosed with PTSD after a tornado. I recommend moving this information up to the section where PTSD is discussed.

Line 144. The line "the Generalized Anxitey Disorder (GAD-7) used to evaluate anxiety" seems out of place.

Line 156: "were more likely to be included criteria of diagnosis as a combination of major depressive episode and SUD." This phrasing is unclear. 

Line 175: Is PTSS "post-traumatic stress syndrome"?

Line 182: "females with low levels of education, and lower household income were more vulnerable to tornado events." Were there statistical analyses of all studies conducted to reach this finding? Or is it a finding from a particular set of studies or a single study? The same applies to the finding that "as social support decreased, the effect of tornado exposure on PTSD and depression in boys increased."

Line 188: is the finding "older adolescents were more likely to report symptoms of PTSD" also supported by reference number 15?

Line: 206: what is meant by "loss of individual characteristics"?

Line 232-234: Unless this was something pointed out by the study's authors, I suggest leaving this to the discussion. If the authors pointed it out, it would need rephrasing to indicate this was the case.

Line 241: Was fear and hope related to individual religiosity? Or was the finding about individual religiosity and physical behavior from a different study?

Line 245: The phrasing "as we expected" may be more suited for the discussion than the results.

Line 246: I suggest rephrasing to "following tornadoes among the respondents of two studies"

Line 253: I suggest rephrasing to "According to one study, the impact of PTSD symptoms an adolescent alcohol use outcomes can be mitigated"

Line 255: What is meant by "genetic tendency to consume alcohol"?

Lines 277-279 are more suited for the discussion than the results section. 

Line 329. What do the authors mean by "a low-substance environment has been shown to reduce genetic risk"?

Line 387: please, do indeed "[include studies from above]." 

Line 396: the line "by applying a wide spectrum such as" seems incomplete.

Line 398: Please delete one of the first "also"

General note: I recommend using sub-headings or other ways to signal the presence of different sections (bold, underline). 

Kind regards, 

Reviewer

Round 2

Reviewer 1 Report

Reviewer report 7 October 2022

Thank you for the opportunity to review this manuscript again. The manuscript appears much improved. A few outstanding points are listed below.

Abstract

·      Lines 10-11 - two different numbers are given for the number of articles retrieved from the search (383 and 384). Please clarify this.

Methods

·      Line 68 – extra words (“related to”) can be deleted in point i)

Results

·      Line 122 - still refers to the “Missouri and Alabama tornado in 2011” when it should read “Missouri and Alabama tornado outbreak in 2011” in line with the revisions made in the discussion section

·      Line 191 – PTSS stands for “post-traumatic stress syndrome,” not “post-traumatic stress symptoms.” Please amend.

·      Table S2 - Authors’ names are inconsistently reported in the table – some report first names first, and others report last names first. Please be consistent throughout the table.

Discussion

·      Lines 320-21 – sentence is incomplete

·      Lines 354-7 – sentences are unclear. Please clarify

·      Lines 369-70 – the review also looked at literature that assessed effects of tornadoes on mental health during tornadoes, as stated in the Methods section. Please add this in.

Author Response

Authors Second Responses to Reviewer Comments (R2)

Thank you for the opportunity to revise and resubmit our manuscript, “Mental health impacts of tornadoes: A systematic review” for publication consideration in International Journal of Environmental Research and Public Health.

We greatly appreciate the reviewer feedback and suggestions for improving the manuscript. Below we have provided responses to each of the reviewer comments.

Again, thank you for this opportunity. The reviewers provided valuable input which has resulted in changes that have strengthened the final paper.

Reviewer comments

Response to reviewers

Abstract

Lines 10-11 - two different numbers are given for the number of articles retrieved from the search (383 and 384). Please clarify this.

Thank you for catching this typo, we have changed this 384.

Methods

Line 68 – extra words (“related to”) can be deleted in point i)

We deleted “related to”.

Results

Line 122 - still refers to the “Missouri and Alabama tornado in 2011” when it should read “Missouri and Alabama tornado outbreak in 2011” in line with the revisions made in the discussion section

We changed it to read “the Missouri and Alabama tornado outbreak in 2011”

Line 191 – PTSS stands for “post-traumatic stress syndrome,” not “post-traumatic stress symptoms.” Please amend.

We have changed it to “post-traumatic stress syndrome”.

Table S2 - Authors’ names are inconsistently reported in the table – some report first names first, and others report last names first. Please be consistent throughout the table.

Thank you, we revised it. It is consistent throughout the table now.

Discussion

Lines 320-21 – sentence is incomplete

Thank you, we deleted “and we recommend”.

Lines 354-7 – sentences are unclear. Please clarify

We have rewritten this section of the discussion and we think it is now more clear. Please see lines 357-369

Lines 369-70 – the review also looked at literature that assessed effects of tornadoes on mental health during tornadoes, as stated in the Methods section. Please add this in.

Thank you for the feedback, we rephrased it “In this review, we summarized peer-reviewed literature examining mental health impacts during and after tornado events.”

Round 3

Reviewer 1 Report

Thank you for the revised and improved version of the manuscript. No further revisions are suggested.